# Descriptive Analysis of Common Fusion Mutations in Papillary Thyroid Carcinoma in Hungary

**DOI:** 10.3390/ijms251910787

**Published:** 2024-10-08

**Authors:** Richard Armos, Bence Bojtor, Janos Podani, Ildiko Illyes, Bernadett Balla, Zsuzsanna Putz, Andras Kiss, Andrea Kohanka, Erika Toth, Istvan Takacs, Janos P. Kosa, Peter Lakatos

**Affiliations:** 1Department of Medicine and Oncology, Faculty of Medicine, Semmelweis University, 1083 Budapest, Hungary; armos.richard@semmelweis.hu (R.A.); bojtorbence01@gmail.com (B.B.); putz.zsuzsanna@med.semmelweis-univ.hu (Z.P.); takacs.istvan@semmelweis.hu (I.T.); kosa.janos@semmelweis.hu (J.P.K.); 2SE HUN-REN-TKI ENDOMOLPAT Research Group, 1085 Budapest, Hungary; balladetti@gmail.com; 3Department of Plant Systematics, Ecology and Theoretical Biology, Eötvös Loránd University, 1117 Budapest, Hungary; janos.podani@ttk.elte.hu; 4Department of Pathology, Forensic and Insurance Medicine, Faculty of Medicine, Semmelweis University, 1091 Budapest, Hungary; illyes.ildiko@semmelweis.hu (I.I.); kiss.andras@semmelweis.hu (A.K.); 5Department of Surgical and Molecular Pathology, National Institute of Oncology, 1122 Budapest, Hungary; kohanka.andi@gmail.com (A.K.); dr.toth.erika@oncol.hu (E.T.)

**Keywords:** papillary thyroid carcinoma, thyroid cancer, fusion mutations, gene fusions, clinicopathological associations, molecular diagnostics

## Abstract

Thyroid cancer is the most common type of endocrine malignancy. Papillary thyroid carcinoma (PTC) is its predominant subtype, which is responsible for the vast majority of cases. It is true that PTC is a malignant tumor with a very good prognosis due to effective primary therapeutic approaches such as thyroidectomy and radioiodine (RAI) therapy. However, we are often required to indicate second-line treatments to eradicate the tumor properly. In these scenarios, molecular therapies are promising alternatives, especially if specifically targetable mutations are present. Many of these targetable gene alterations originate from gene fusions, which can be found using molecular diagnostics like next-generation sequencing (NGS). Nonetheless, molecular profiling is far from being a routine procedure in the initial phase of PTC diagnostics. As a result, the mutation status, except for *BRAF V600E* mutation, is not included in risk classification algorithms either. This study aims to provide a comprehensive analysis of fusion mutations in PTC and their associations with clinicopathological variables in order to underscore certain clinical settings when molecular diagnostics should be considered earlier, and to demonstrate yet unknown molecular–clinicopathological connections. We conducted a retrospective fusion mutation screening in formalin-fixed paraffin-embedded (FFPE) PTC tissue samples of 100 patients. After quality evaluation by an expert pathologist, RNA isolation was performed, and then NGS was applied to detect 23 relevant gene fusions in the tumor samples. Clinicopathological data were collected from medical and histological records. To obtain the most associations from the multivariate dataset, we used the *d*-correlation method for our principal component analysis (PCA). Further statistical analyses, including Chi-square tests and logistic regressions, were performed to identify additional significant correlations within certain subsets of the data. Fusion mutations were identified in 27% of the PTC samples, involving nine distinct genes: *RET*, *NTRK3*, *CCDC6*, *ETV6*, *MET*, *ALK*, *NCOA4*, *EML4*, and *SQSTM1*. *RET* and *CCDC6* fusions were associated with type of thyroidectomy, RAI therapy, smaller tumor size, and history of Hashimoto’s disease. *NCOA4* fusion correlated with sex, multifocality, microcarcinoma character, history of goiter, and obstructive pulmonary disease. *EML4* fusion was also linked with surgical procedure type and smaller tumor size, as well as the history of hypothyroidism. *SQSTM1* fusion was associated with multifocality and a medical history of thyroid/parathyroid adenoma. *NTRK3* and *ETV6* fusions showed significant associations with Hashimoto’s disease, and *ETV6,* also with endometriosis. Moreover, fusion mutations were linked to younger age at the time of diagnosis, particularly the fusion of *ETV6*. The frequent occurrence of fusion mutations and their associations with certain clinicopathological metrics highlight the importance of integrating molecular profiling into routine PTC management. Early detection of fusion mutations can inform surgical decisions and therapeutic strategies, potentially improving clinical outcomes.

## 1. Introduction

Thyroid cancer is the most common type of endocrine cancer, with above 800,000 newly registered cases in 2022 globally [1,2]. Its incidence has more than doubled in the last 30 years in the USA [3,4]. About 90% of the patients diagnosed with thyroid cancer have its differentiated (DTC) subtype, including, basically, the papillary (PTC) and the follicular (FTC) histological variants [5]. For better therapeutic outcomes, the widely known conventional histological categorization of thyroid cancer and its subtypes is increasingly required to be further classified into molecular categories as well [6].

The primary choice of treatment for thyroid cancer, if technically feasible, is its surgical removal by partial or total thyroidectomy, depending on the risk of recurrence [7,8]. Based on the American Thyroid Association (ATA) guidelines, regular risk assessment is recommended during postoperative management to estimate therapeutic response and prognosis and to prevent relapse. Risk assessment should take factors like histological type, extrathyroidal invasion, (lympho)vascular invasion, presence of distant metastases, and postoperative serum marker (thyroglobulin, anti-thyroglobulin) levels into consideration. Prognostic value is increasingly associated with the molecular profile of the cancer as well [7,9,10,11]. If the cancer falls into the intermediate-risk or high-risk category and contraindication is not present, post-surgical adjuvant I^131^ radioiodine (RAI) is recommended as part of the first-line treatments. RAI can be repeated several times if molecular or structural recurrence happens [7,12].

In case of therapy-refractory cancer or loss of radioiodine uptake, molecular target therapies can be considered as second-line options depending on the clinical setting [7,12,13,14]. ATA guidelines recommend active surveillance for metastatic DTCs without symptoms with a diameter between 1 and 2 cm and an active intervention if the tumor growth rate increases. Additionally, molecular therapies can be indicated in metastatic DTC cases [12]. Molecular therapies taking effect on the mitogen-activated protein kinase (MAPK) signaling pathway can sometimes even reinduce radioiodine sensitivity in tumors previously proved to be refractory to RAI [7,15]. Besides those related to MAPK, other significant thyroid cancer signaling pathways include molecules such as phosphatidylinositol 3-kinase (PI3K) and phosphatidylinositol phospholipase C (PLCγ) [16,17]. As molecular therapies are usually highly specific, indicating them as second-line approaches can only happen after successfully detecting a targetable mutation within the tumor. Selective molecular therapies are associated with greater efficacy and a more tolerable side effect profile [18]. However, in cases where a specifically targetable mutation cannot be found (e.g., *RET*), the usage of anti-angiogenic multikinase inhibitors (AAMKIs) might be a viable alternative [15]. In general, if first-line approaches fail to achieve the therapeutic goal, and a selectively targetable mutation can be identified while contraindications are not present, application of the appropriate molecular therapy is the choice of treatment; except for the *BRAF* inhibitors against *BRAF V600E* mutation in DTC. These usually come only after failed or contraindicated AAMKI treatments. It is worth mentioning that the RECIST objective response rate (ORR) and the progression-free survival (PFS) of lenvatinib, an AAMKI, was superior compared to the *BRAF*-selective dabrafenib ± trametinib in the SELECT study [7,15].

Detection of several classes of gene and molecular alterations is possible such as hotspot mutations, copy number variations, fusion mutations, and miRNA expression differences [17,19,20]. Gene fusions basically occur when a single hybrid gene is created by the merger of two distinct genes after an improper chromosomal rearrangement, which causes chromosome segments to break and then reattach incorrectly. Defects of this kind can act as driver mutations and facilitate cancer development. A classic example of such a mutation is the Philadelphia chromosome created by the translocation of the *BCR* and *ABL1* genes between chromosomes 9 and 22, leading to chronic myeloid leukemia [21]. However, the oncogenic effect of gene fusions is evident in the case of solid tumors, like PTC as well [17]. The clinically most relevant, frequently occurring and selectively targetable, gene alterations in thyroid cancer are the different types of *BRAF*, *RET*, *NTRK*, and *MET* mutations [19,22,23,24,25,26,27]. An aberrant *RET* gene is quite frequent not only in medullary thyroid cancer but in DTC (~10%) as well [19,28,29]. Selective *RET* kinase inhibitors targeting the signaling pathway of *RET* are available to use in such a mutation pattern. They have tolerable side effects and their efficacy, regarding paraneoplastic symptoms as well, is explicit.

Members of the neurotrophic tyrosine receptor kinase (*NTRK*) gene family (*NTRK1*, *NTRK2*, *NTRK3*) can also often carry somatic mutations. In these cases, potential oncogenicity is mainly linked to protein fusions of tyrosine receptor kinases on thyrocytes and their autonomous MAPK, PLCγ, and PI3K signaling activity, consequently. Potent second-line treatments in DTC counteracting the effects of *NTRK* fusions are available as well. These medications showed great therapeutic response rates accompanied by few side effects. In addition, they showed the ability to help tumor tissue regain its previously lost radioiodine uptake ability. The clinical value of inhibiting the molecular pathway related to *MET* in DTC is currently under investigation in clinical trials [13,15,16,25,26,27,29].

Fusion mutations can be detected via multiple methods such as polymerase chain reaction (PCR), fluorescence in situ hybridization (FISH), immunohistochemistry (IHC), and next-generation sequencing (NGS). These molecular methods can be applied on fresh tissues and also on fine-needle aspiration biopsy (FNAB) and formalin-fixed paraffin-embedded (FFPE) specimens [17,21,22,23,29,30,31].

It is true that the role and importance of fusion mutations in PTC are quite well understood; however, original molecular studies evaluating their connection to everyday clinical and histological metrics related to PTC are still much needed. With this study, we would like to provide a comprehensive analysis of the distribution of fusion mutations in a heterogenous PTC cohort in Hungary, as well as to determine fusion mutation–clinicopathological metric associations in order to point out the potential of molecular diagnostics in PTC if used more frequently, even in earlier stages or in special clinical scenarios. Therefore, we conducted a retrograde fusion mutation screening in the context of 23 relevant gene fusions in the tumor samples related to 100 consecutive PTC patients applying NGS. Then, correlations among molecular and clinicopathological variables of the PTC cases were calculated to find any underlying associations between them.

## 2. Results

### 2.1. Study Population

RNAs were isolated from tissue samples related to 107 consecutive patients previously diagnosed with PTC and found to be eligible for our study. Later, 7 samples were excluded due to inapplicable isolate or incomplete related clinical data resulting in a total of 100 samples with molecular results suitable for further investigation. The study population included considerably more women (*n* = 71) than men (*n* = 29). The mean age (±SD) at diagnosis was 45 years (±15.64 years; women: 46 ± 16.15 years; men: 44 ± 14.52 years). Thirty-two patients did not have any comorbidities, except PTC. Sequencing was applied to detect 23 different types of highly relevant fusion mutations in the context of PTC pathogenesis. Samples analyzed in our study included the PTC subtypes shown in Table 1.

### 2.2. Descriptive Analysis

Sequencing data revealed the distribution of the fusion mutations of interest in the PTC cohort, offering a detailed view of the underlying molecular pattern of the disease development as represented in Figure 1. We were able to identify some types of fusion mutation in 27% of the samples. The following nine genes were found to be affected by fusion mutations: RET (28.57%), NTRK3 (16.33%), CCDC6 (16.33%), ETV6 (12.24%), MET (8.16%), ALK (4.08%), NCOA4 (8.16%), EML4 (4.08%), and SQSTM1 (2.04%). The most notable malignancies, with sufficient data in the literature, associated with the fusion genes identified in this study are listed in Table 2 [32,33,34,35].

### 2.3. Principal Component Analysis (PCA) Based on d-Correlation of Mixed Scale-Type Variables of Fusion Mutation Status and Clinicopathological Data

Deploying PCA (Figure 2) with d-correlation made it possible to analyze multiple variable types, such as nominal, ordinal, interval, and ratio-scale, simultaneously [36]. Each point on the PCA plot is labeled to indicate specific clinicopathological variables or fusion mutation status. Our method helps visualize larger groups of variable types and their distribution relative to each other. On the left on Axis 1, it is clearly visible that therapy-related variables clustered well with prognostic ones and with those associated with worse clinical outcomes in general [7]. However, all fusion mutation-related variables are arranged on the far right side of Axis 1, most of them encircled by multiple clinicopathological variables. The opposite localization of gene fusions from therapy-related and prognostics-related variables suggests a rather negative correlation between them. Common fusion partner genes such as RET/CCDC6 and NTRK3/ETV6 clustered well with each other appropriately [16,17,19,23].

Carrying a fusion mutation within PTC tissue might be linked to a previous diagnosis of Hashimoto’s disease or endometriosis as well as to the accumulation of thyroid diseases in the family. In the cases of RET, CCDC6, MET, EML4, and ALK, a similar clustering was observed with each other and the type of thyroidectomy performed (near-total thyroidectomy, lobectomy, partial surgery with or without completion), the indication of radioiodine (RAI) therapy, a smaller than 1 cm tumor size in diameter during preoperative diagnostics with imaging techniques, a medical history of Hashimoto’s disease and hypothyroidism, as well as a family history of any thyroid-related disease. Furthermore, clustering suggests a link between SQSTM1 fusion and a medical history of thyroid/parathyroid adenoma, and features such as multifocality and sidedness on histology; links were also found between NCOA4 fusion and sex, histological features of two-sidedness, multifocality and microcarcinoma character, a medical history of goiter, and, interestingly, obstructive pulmonary disease. Strangely, the patient’s age, clinical staging (based on the 8th edition of the American Joint Committee on Cancer (AJCC)), and history of malignancies other than thyroid cancer did not exhibit any correlation with the investigated fusion mutation types by the d-correlation method [37].

### 2.4. Deeper Analysis for Associations between NTRK3 and ETV6 Fusions and Clinicopathological Variables

Although it is true that NTRK3 and ETV6 fusions seemingly did not cluster well with clinicopathological variables via d-correlation, we hypothesized that it could be due to the more comprehensive rather than detail-oriented nature of this method compared to conventional statistics. d-correlation is quite effective for large, multivariate datasets, but applying more ordinary methods can be beneficial if analyzing fewer variables. In this respect, a deeper investigation was performed to determine the relation of these two fusion mutations to other variables using other statistical approaches as well. As a result, further potential associations could be revealed.

To evaluate associations between fusion mutation status and other binary-type clinicopathological variables, we first performed a Chi-square test to detect any significant (*p* < 0.05) links. Then, for significant associations, we continued the analysis with a logistic regression model to determine the strength and direction of the relationship. Both NTRK3 and ETV6 fusion mutations showed a marked positive correlation with Hashimoto’s thyroiditis with an approximate 13-fold and 21-fold increased likelihood of their co-occurrence with the disease compared to other cases in the cohort, respectively. In addition, ETV6 fusion was associated with a history of endometriosis as presented in Figure 3. The likelihood of their co-occurrence was more than 15-fold.

Statistical tests were also carried out for non-binary/nominal variables to identify any links with the presence of NTRK3 or ETV6 fusions. Therefore, the following non-binary/nominal variables were studied: histological subtype of PTC, cancer localization within the thyroid, and type of thyroidectomy. Similarly to binary co-variables, we performed a Chi-square test to determine significance. Only the type of thyroidectomy showed significant associations with NTRK3 and ETV6 fusions. The type of PTC surgery was divided into three categories: primary (near-)total thyroidectomy, not-total thyroidectomy (lobectomy, partial resection), and secondary total thyroidectomy (completion after not-total thyroidectomy).

To determine which surgical procedure is the most strongly predicted by the presence of the two tested fusions, we applied a multinomial logistic regression model. A baseline surgical category (not-total thyroidectomy) was defined, and then coefficients were established indicating how changes in the fusion mutation status might have influenced the other surgical procedure categories relative to the baseline category (Figure 4). In both NTRK3 and ETV6 fusion-positive PTC patient groups, total thyroidectomy was performed the most often. Interestingly, though, more patients needed secondary completion surgery than those who had their entire thyroid removed in the first instance. NTRK3 and ETV6 gene fusions were accompanied by an approximately 15–33% increased likelihood of needing a total thyroidectomy at some point during the course of patient care, with the need for a secondary completion surgery being the most pronounced. These results highlight the potential of preoperative genetic testing, at least for carriers of these mutations, for better planning of the surgical approach in PTC and to avoid the unnecessary repetition of surgeries.

### 2.5. Analysis for Associations between Ratio-Scale, Ordinal, and Clinicopathological Variables

The ratio-scale variables were first checked for normality. Only one variable (age at diagnosis) followed the normal distribution; the other ratio-type variables were not normally distributed.

After the normality test, we evaluated the homogeneity of variances in the context of those fusion-related variables occurring at least six times or more in the cohort. Next, significant differences in mean age between fusion-positive and fusion-negative patients were determined by applying an independent two-sample *t*-test for most of the fusion-positive cases and a Welch’s *t*-test in the case of ETV6 fusion due to potential violation of the homogeneity of variances assumption. Compared to the fusion-negative cases, a significant difference was observed in the mean age of patients carrying fusion mutations of NTRK3 and ETV6. These patients were much younger when diagnosed with PTC (mean age: 35.4 and 32.0 years, respectively) than those without any gene fusions (mean age: 47.8 years), as illustrated in Figure 5. In addition, patients’ age with a generally positive status for fusion mutations (mean age: 39.0 years) showed a significant difference from the fusion-negative patients’ age, suggesting a similar impact on the time of onset of PTC by other, less frequently occurring, therefore not individually included, fusion mutations as well.

Significant associations could not be detected with ordinal variables (clinical stages based on AJCC 8th edition, TNM stages, R stage, and ATA risk score) or with not-normally distributed ratio-scale variables (tumor size under microscope, cumulative radioiodine dose, cumulative external beam radiation therapy dose).

### 2.6. Outlook toward Not Fusion-Related Clinicopathological Associations of PTC Patients

Variables followed our clinical expectations in general; smaller PTCs underwent relapse fewer times than bigger ones, PTCs with more advanced histological features relapsed more frequently than their counterparts, etc. Interestingly, though, molecular therapies were usually indicated in a more advanced disease state of being after a relapse, showing thyroid capsule-, extrathyroidal-, and/or lymphovascular invasion, or with the need for external beam radiation therapy (EBRT) as represented in Figure 6. This observation greatly contrasts the fact that fusion mutations, which are frequent targets of these therapies, showed rather negative correlations with the same clinicopathological variables via the d-correlation method. This may be a sign that these patients would benefit from earlier molecular diagnostics during their course of medical management and an earlier initiation of molecular target therapies if possible.

## 3. Discussion

In our comprehensive study on the relationship between gene fusions and clinicopathological status in the PTC cohort, we found that almost one-third (27%) of the PTC patients carried a fusion mutation within their tumor tissue. In the 2014 TCGA study, involving 484 individual PTC cases, fusion mutations occurred in 15.3% of the cases [19]. Another study from 2017 detected fusion mutations in only 7.97% of the PTC cases [24]. However, a recent study discovered fusion mutations in 29.86% of advanced DTC cases, which is quite similar to our results [22]. The finding that *RET*-related fusion mutations are the most common in our study is consistent with the recent literature data [23]. The frequent occurrence of such mutations highlights the relevance of screening for them more often in everyday clinical practice. This is amplified by the fact that the fusion proteins originating from the majority of the detected driver mutations, namely those of *RET*, *NTRK3*, *CCDC6* (when co-occurs with *RET*), and *MET*, can be effectively targeted with small tyrosine kinase inhibitors in thyroid cancer [25,26,27].

The most optimal way to illustrate the relation between all examined measures and gene fusion-related variables seemed to be the application of PCA. As the study involved multiple variable types, comparing them simultaneously was a relatively complex task and required a special approach. Therefore, we decided to use a novel statistical framework, named *d*-correlation, which calculates the matrix correlation based on semi-matrices derived for all pairs of observations [36]. *d*-correlations showed us that the observed gene fusions mainly cluster with each other but also with some other clinicopathological variables. PTC patients with comorbidities like endometriosis or Hashimoto’s thyroiditis or those with a family history of any thyroid disease have a greater chance of a positive fusion mutation status in general. Thyroid disease in the patient’s family, Hashimoto’s disease, and hypothyroidism, as well as the type of thyroidectomy, the need for indicating RAI, and smaller tumor size, clustered quite well with gene fusions related to *RET*, *CCDC6*, *MET*, *EML4*, and *ALK*. Moreover, the patient’s sex, comorbidities such as goiter and obstructive pulmonary disease, or histological features like microcarcinoma, two-sidedness, and multifocality tended to cluster with the *NCOA4* fusion gene. *SQSTM1* fusion also clustered well with multifocality and with a medical history of thyroid/parathyroid adenoma.

Interestingly, *d*-correlation did not reveal any marked associations with clinicopathological variables in the context of *NTRK3* and *ETV6* fusions, which, on the other hand, clustered very well together. This is consistent with the observation that *NTRK3/ETV6* fusion pairs were particularly frequent in the cohort. Due to this discrepancy, we would have liked to investigate the correlation pattern of these two fusion genes separately, as well as apply more detailed, conventional statistical methods this time. As a result of these methods, we identified some additional associations both in the case of *NTRK3* and *ETV6*. PTC patients with a history of Hashimoto’s disease showed a positive correlation of having *NTRK3* and/or *ETV6* fusion mutations as well. Additionally, *ETV6* positively correlated with the medical history of endometriosis. Moreover, total thyroidectomy was significantly more often indicated than not-total thyroidectomy in groups having *NTRK3* and/or *ETV6* fusions. Moreover, compared to the number of primary total thyroidectomies, significantly more patients needed a secondary completion of a subtotal thyroidectomy with these mutation statuses. This means that *NTRK3* and *ETV6* fusions might be causally related to the extent of the tumor mass, and a primary total thyroidectomy might be more beneficial over a subtotal one for those patients having either *NTRK3* or *ETV6* fusion mutation within the PTC tissue. This could help prevent secondary surgeries.

We also found that the patients’ age at diagnosis was much younger in those PTC cases with detected fusion mutations relative to those without any fusions. This observation was proven to be significant in the context of *NTRK3* and *ETV6* fusions and fusion mutation positivity in general, suggesting the importance of molecular diagnostics in younger-than-average PTC patients. This finding, suggesting that fusion mutation frequency is age-dependent in PTC, is concordant with previous results in the literature [22,38]. Additionally, fusion mutations are more commonly found in pediatric patients; moreover, these mutations are associated with a younger age in adults, as well as in pediatric PTC patients [22,38]. It is important to note that the average age in our whole cohort was consistent with the literature data [39].

The association between clinicopathological features and fusion mutations has been previously studied in PTC. For instance, a metanalysis showed that *NTRK3*-fused PTC cases had an increase in disease aggressiveness and shorter progression-free survival (PFS) when compared to *NTRK1*-fused PTC cases [40]. The possible associations between fusion mutations and radioactive iodine (RAI) refractoriness have also been studied previously [22,41]. Another study also reported that *RET*-rearranged tumors are more likely to have an advanced disease state compared to *BRAF*-mutant and *RAS*-mutant PTC cases [42].

As a side analysis, we also evaluated the relation of clinicopathological variables relative to each other without taking fusion mutations themselves into account. In this respect, we analyzed 35 different variables commonly documented when managing PTC patients [7,37]. Most of the associations discovered were unsurprising from a clinical point of view. However, the indication of molecular therapies showed a positive correlation with variables generally linked to a more advanced cancer, like the need for EBRT as well as the tendency of relapse, invasion of the thyroid capsule, the extrathyroidal space, or the surrounding small vessels. Despite this, advanced disease-related variables correlated negatively with the targetable fusion mutations in our study. Contrary to the literature, this indicates that molecular therapies might have a role in earlier stages of the disease, and reserving them only for advanced scenarios might not benefit the patients overall since the molecular targets of most of these treatments prefer to occur in seemingly more peaceful PTCs [15].

The initial diagnosis and treatment options for the disease are already highly effective, with low rates of recurrence and complications. However, there is still the potential for further improvement in the patients’ follow-up [43]. PTC management relies strongly on risk evaluations, such as the ATA risk stratification system. The most recent version of this system already considers the *BRAF V600E* mutation when calculating the likelihood of disease recurrence [7]. On the other hand, it does not discuss the significance of other mutations. Our study highlights that gene fusions correlate rather negatively with the ATA risk score and most of the classical metrics used to calculate it. In fact, having a targetable mutation is clearly accompanied by the advantage that additional therapeutic options are available for the particular patient. The involvement of gene fusion status, especially those fusions with accessible target therapies, in the risk stratification protocol could improve its accuracy.

## 4. Materials and Methods

### 4.1. Study Population, Sample Collection, and Histopathological Processing

PTC tissue samples originating from previous thyroid surgical material related to a total of 100 consecutive and anonymized cases related to the Department of Medicine and Oncology, Semmelweis University, Hungary, and the National Institute of Oncology, Hungary, were involved in the study. Anonymized medical and pathological data were queried from clinical databases, while tissue blocks were collected from histopathological archives of the Department of Pathology, Forensic and Insurance Medicine, Semmelweis University, Hungary.

In our investigation, we evaluated the following clinicopathological features of the cases: sex, age at diagnosis, histological subclassess of PTC, aggressiveness of the PTC variant, clinical stage based on the 8th edition of the AJCC, TNM stage, R stage, ATA risk score, tumor size, lymphovascular invasion, perineural spread, capsule invasion, extrathyroidal spread, focality, microcarcinoma character, cancer localization and sidedness within the thyroid lobes, preoperative imaging features (tumor size, sidedness, infiltrative character, lymph node involvement, presence of distant metastasis), type of surgical procedure performed (including lymph node dissection), features of indicated treatments (radioiodine, external beam radiation therapy (EBRT), molecular therapy, relapse after treatment (molecular and/or structural), complication after treatment), data from medical history (family history of thyroid disease, prior chemotherapy, Hashimoto’s disease, prior hyperthyroidism or hypothyroidism, prior goiter, association with thyroid or parathyroid adenoma, and association with comorbidity clusters such as other malignancy, breast cancer, benign tumor, uterine myoma, diabetes, cardiovascular morbidity, respiratory disease, gastric acid related disorder, appendicitis, autoimmune disease, gallstones, musculoskeletal disorder, kidney stones, endometriosis).

Investigated tissue samples were previously preserved within formalin-fixed paraffin-embedded (FFPE) blocks from which hematoxylin–eosin-stained probe sections were produced to confirm the sufficient amount of tumor presence and its expected histological type, and to determine the exact percentage of the tumor volume. This was performed by a qualified pathologist. We carried out dissection of the pre-selected tissue blocks resulting in 10 µm thick curls (4 pieces per block). The obtained tissue curls were then forwarded to molecular analysis.

### 4.2. Molecular Processing for Identification of Fusion Mutations (RNA Isolation, Quality Control (QC), RNA Quantification and Sequencing)

The RecoverAll™ Total Nucleic Acid Isolation Kit (Life Technologies, Carlsbad, CA, USA) was used for the isolation of DNA-free RNA. In short, the paraffin was removed with xylene and ethanol treatment. The pellets were digested with proteinase K solution in heat blocks for 15 min at 50 °C, then 15 min at 80 °C. Samples were combined with Isolation Additive, ethanol, and then RNA were captured by the column. After several washes and incubation with the presence of DNAse, purified RNA was eluted into a 60 μL elution buffer. The concentration of the isolated RNA was measured with the Qubit RNA HS Assay Kit (Life Technologies, Carlsbad, CA, USA).

The Oncomine Focus amplicon library was prepared using the Ion AmpliSeq Library Kit 2.0 (Life Technologies, CA, USA); briefly, multiplex primer pools were added to 10 ng of genomic DNA, and after reverse transcription, to 10 ng of total RNA, then amplified with the following PCR cycles: at 99 °C for 2 min, at 99 °C for 15 s, and at 60 °C for 4 min (23 cycles), and holding on at 10 °C. Primers were partially digested using a FuPa reagent, then sequencing adapters were ligated to the amplicons. Agencourt AMPure XP Reagent (Beckmann Coulter, CA, USA) was selected for library purification. The concentration of the final library was determined by the qPCR method run by the QuantStudio instrument (Life Technologies, CA, USA).

Template preparation was performed with the Ion OneTouch kit (Life Technologies, CA, USA) on a semiautomated Ion OneTouch instrument using an emPCR method. After breaking the emulsion, the nontemplated beads were removed from the solution during the semiautomated enrichment process on an Ion OneTouch ES (Life Technologies, CA, USA) machine. After adding the sequencing primer and enzyme, the Ion Sphere Particle (ISP) beads were loaded into an Ion 520 sequencing chip, and the sequencing runs were performed using the Ion S5 Sequencing kit (Life Technologies, CA, USA) with 500 flows.

### 4.3. Data Analysis via Bioinformatics and Statistical Evaluation

In the PTC tumor samples, we aimed to detect the presence or absence of the following 23 different oncogenic driver gene fusion mutations: *ABL1*, *AKT3*, *ALK*, *AXL*, *BRAF*, *EGFR*, *ERBB2*, *ERG*, *ETV1*, *ETV4*, *ETV5*, *FGFR1*, *FGFR2*, *FGFR3*, *MET*, *NTRK1*, *NTRK2*, *NTRK3*, *PDGFRA*, *PPARG*, *RAF1*, *RET*, *ROS1*.

To explore the distribution of the listed fusion gene partners compared to other gene partners, other fusion mutations, and fusion-negative cases, variant annotation and cloud-based data analysis were performed using Ion Reporter 5.18 platform (Life Technologies, CA, USA) with pre-defined parameters followed by descriptive statistics.

After successfully collecting molecular data, we complemented the clinicopathological dataset with sequencing information regarding the fusion mutation status for each case. Due to the large number of variables and the heterogeneity of the variable types within the clinicopathological dataset, which contains nominal, ordinal, and ratio-scale types, we applied multiple statistical tests to improve the quality of our multivariate analysis.

Descriptive statistics of basic clinical data were performed using IBM SPSS 27.0 (SPSS Inc., Chicago, IL, USA).

Association mapping within the entire dataset (including the listed fusion mutations and all the clinicopathological data) could not have been achieved by conventional multivariate techniques. The reason is that the variables are heterogeneous in terms of measurement scale. Fusion mutation-related variables are binary nominal as well as many other clinicopathological variables (e.g., aggressiveness, multifocality, relapse). However, other variables belonged to ordinal (e.g., clinical stage, TNM stage, ATA risk score) and ratio (e.g., age, tumor diameter) types. In case of nominal variables (or binary/nominal variables), only the equality or non-equality of character states can be established. Arithmetic variances only make sense in the case of ratio-type variables, while for ordinal variables, only the sequence of states can be interpreted. Therefore, special methods need to be used to uncover variable associations in the most effective way possible. A new method, the *d*-correlation formula, recently published by Podani et al., was applied to determine correlations across mixed-type variables by reducing the number of dimensions of the dataset. Then, the comprehensive correlation matrix, which was generated by this formula, could be visualized with principal component analysis (PCA).

For specific subsets of data, we also performed more conventional statistical analyses to extend the scope of the investigation to potential correlations that were not apparent from PCA. For this, we mainly focused on associations of *ETV6* and *NTRK3* partner genes as they were seemingly not clustering well with other variables via PCA. Moreover, we assessed the relation of fusion mutation status (including general fusion mutation positivity and those specific fusions occurring at least six times in the dataset) to the patients’ age at diagnosis, and the association pattern exclusively between not fusion-related, binary-type clinicopathological variables. For the reasons detailed above, we were limited in terms of effectively comparing multiple variable types with each other in such a way. This statistical analysis was conducted using the Python v3.8 programming environment. The significance of associations between fusion mutations (binary/nominal) and other nominal-, and ordinal-type clinicopathological variables was calculated by the Chi-square test (*p* < 0.05) using the SciPy v1.7.3 package. Associations found to be significant from these subsets of data were then further evaluated with logistic regression models to determine the strength and direction of the correlations, for binary variables exploiting the scikit-learn v1.2.2 package. We used multinomial logistic regression for not-binary nominal variables, as it handled multiple different categories of the same variable better. For ordinal variables, we applied the Ordered Logit model using the Statsmodels package v0.13.2.

For ratio-type variables, we first checked the normal distribution of the values. Only age at diagnosis showed a normal distribution, while the diameter of the tumor, the cumulative dose of RAI therapy, and the cumulative dose of EBRT were not normally distributed. Given the size of the dataset, the evaluation of normal distribution was performed by the Shapiro–Wilk test. The normally distributed variable (age at diagnosis) was further tested for the homogeneity of variances. Next, the mean age at diagnosis was investigated in the context of fusion mutation status by applying an independent two-sample *t*-test or, in the case of *ETV6* fusion, a Welch’s *t*-test due to potential violation of the homogeneity of variances assumption. With the *t*-tests, we were able to determine any significant differences between the mean age of the fusion-positive and fusion-negative groups of patients. In cases of not normally distributed variables, the Kruskal–Wallis test was used to determine any significant associations to fusion mutations. As no significant association pairs could be identified, we did not continue with a deeper analysis in this direction. For statistical evaluations of ratio-type variables, we applied the SciPy package as well.

### 4.4. Literature Search

The literature review was performed using NCBI’s PubMed database to identify disease associations and biological interactions related to fusion mutations, ensuring the inclusion of the most recent studies available at the time of access. Our goal was to select peer-reviewed articles that contained relevant information about the specific gene fusions we were studying. The literature search was conducted up to 31 August 2024. A variety of synonymous search terms were simultaneously employed to gather all the required information from the existing literature. These search terms included “papillary thyroid carcinoma”, “PTC”, “thyroid carcinoma”, “fusion mutation”, “gene fusion” as well as the names of individual genes.

## 5. Conclusions

Our study provides valuable insights into the prevalence and distribution of fusion mutations in PTC in Hungary. Through sequencing of 100 PTC-related FFPE thyroid tissue samples in search of 23 different gene fusions, mutations were identified in 27% of the cases, highlighting the significant role these genetic alterations might play in the pathogenesis of the disease. The most commonly affected driver genes were *RET* and *NTRK3*. The detection of nine distinct fusion genes emphasizes the relatively narrow mutation spectrum of PTC with regard to fusion mutations.

The correlation of these molecular findings with certain clinicopathological variables underscores the value of integrating genetic profiling into routine thyroid cancer diagnostics. PCA powered by the *d*-correlation method revealed specific gene fusion associations with clinicopathological characteristics [36]. For instance, *RET* and *CCDC6* fusions clustered with variables such as type of thyroidectomy, the need for RAI therapy, smaller tumor size, Hashimoto’s disease, and hypothyroidism. *MET*, *EML4*, and *ALK* fusions also clustered with similar variables, along with the family history of thyroid diseases in general. *NCOA4* and *SQSTM1*, however, showed a quite different association pattern. *NCOA4* fusion was associated with patients’ sex, multifocality, microcarcinoma character, medical history of goiter, and obstructive pulmonary disease, while *SQSTM1* fusion was linked with multifocality and medical history of thyroid/parathyroid adenoma. Further, the more conventional statistical analysis identified significant associations of *NTRK3* and *ETV6* fusions with Hashimoto’s disease, plus in the case of *ETV6*, with endometriosis. Both of these fusions were more commonly associated with the need for total thyroidectomy or secondary completion surgeries, suggesting a causal relationship with the indication algorithms of the different surgery types and highlighting the potential benefits of preoperative genetic testing of this kind. Moreover, patients with fusion mutations were diagnosed at a significantly younger age, particularly those with *ETV6* fusions. This underlines the importance of early molecular diagnostics in younger-than-average PTC patients to fine-tune treatment decisions. Interestingly, analysis of clinicopathological–clinicopathological variable pairs raised the possibility that initiating molecular target therapies might be advantageous even in clinically less advanced stages, contrary to the actual clinical practice [15]. Future treatments for PTC are likely to focus on personalized medicine based on the molecular profile of the tumor. This includes expanding the use of targeted therapies tailored to specific genetic mutations found in PTC. Additionally, further combining molecular therapies with standard treatments, such as RAI, may help overcome RAI resistance more frequently, potentially improving treatment efficacy and patient outcomes.

Ultimately, our study helps to understand the underlying connections between the genetic landscape of PTC and the clinicopathological metrics of everyday clinical practice. As a result, it points out the possibility that more widespread molecular diagnostics could enhance diagnostic precision and optimize treatment plans for PTC. To the best of our knowledge, no study in the literature has analyzed the associations between clinicopathological variables and fusion mutations in PTC in such a comprehensive manner. It should be noted, however, that our study has limitations. Hotspot mutations and CNVs were not involved in this study. Additionally, individual cellular pathways related to the detected mutations were not investigated due to the limited amount of samples in the tissue archives suitable for a sufficient quality molecular processing. In addition, our results would be more reliable if repeated measurements had been taken on fusion-positive samples. Moreover, a replacement of FFPE tissue samples with fresh tissues could have improved the quality of sequencing. Compared to similar reports, a larger sample size could have contributed to validating the suspected but not significantly confirmable associations emerging during our study. In addition, this study focuses on a limited number of gene fusions, potentially missing broader genetic alterations explored in other reports. Lastly, patient selection biases might have affected our results as all of our patients were Caucasians living in Eastern–Central Europe.

## Figures and Tables

**Figure 1 ijms-25-10787-f001:**
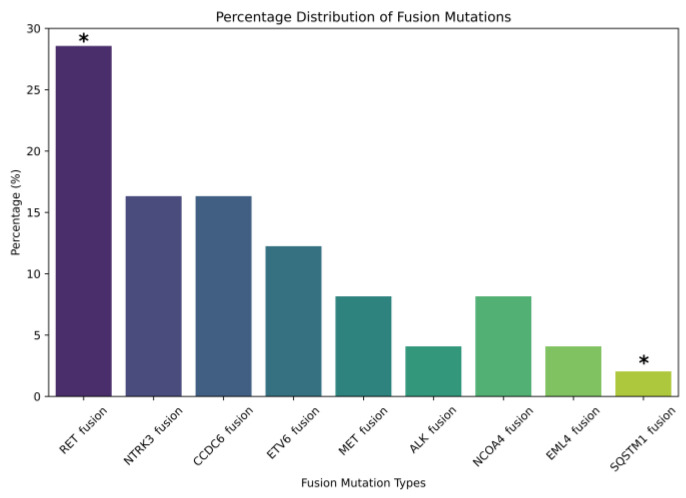
Calculated relative distribution of partner genes associated with detected driver fusion mutations (*n* = 27) in the PTC cohort without representing fusion non-carrier cases (*n* = 73). The relative frequency of occurrence was significantly different (Chi-square test) between *RET* and *SQSTM1* fusions (*p* = 0.026) as marked (*) on the plot. The most frequently identified fusion genes were *RET* (28.57%) and *NTRK3* (16.33%) and their common gene partners *CCDC6* and *ETV6*, respectively.

**Figure 2 ijms-25-10787-f002:**
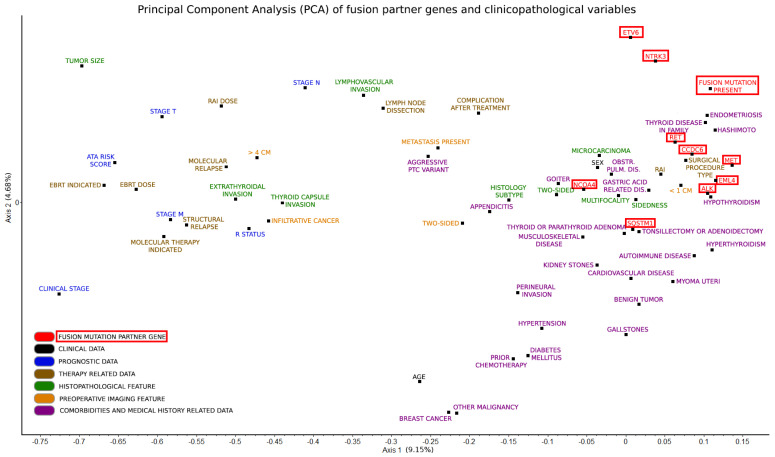
Principal component analysis (PCA) of fusion genes and clinicopathological variables using *d*-correlation for mixed scale types. Black points (variable positions) are labeled and color-coded (bottom left corner) to reflect the grouping of different individual variables into larger categories. Variables related to gene fusion status are indicated with red rectangles. It is well demonstrated that most fusion mutation-related variables tended to cluster with specific clinicopathological variables (middle right side). Therapy-related and prognostics-related variables (middle left side), however, correlated negatively with gene fusions.

**Figure 3 ijms-25-10787-f003:**
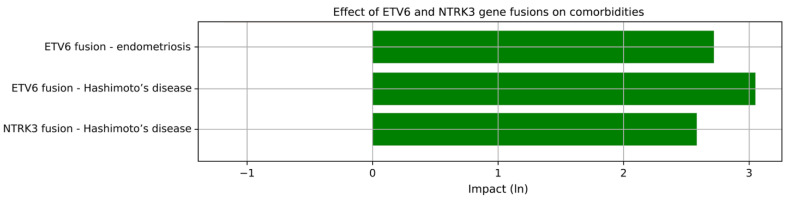
This horizontal bar chart displays the impact of carrying *ETV6* and/or *NTRK3* gene fusions on having certain comorbidities. The values are derived from a logistic regression analysis of the *ETV6* and *NTRK3* gene fusion partners (independent variables) and those binary/nominal-type clinicopathological features (dependent variables) that were associated with these fusions in a significant manner (*p* < 0.05). The length of the bars depends on the strength of the associations relative to other variable constellations in the cohort. All links represented are above the 0 value threshold on the x-axis, indicating that the directions of all the correlations are positive.

**Figure 4 ijms-25-10787-f004:**
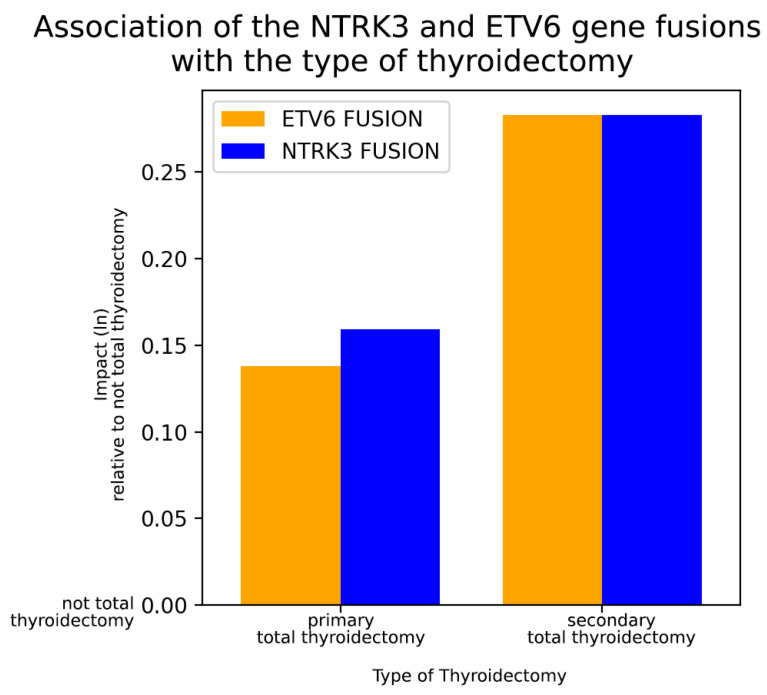
Using Chi-square test (*p* < 0.05), the type of surgical procedure performed was found to be significantly different when *NTRK3* and/or *ETV6* fusions occurred compared to those cases without these fusions. This vertical bar chart of these two significant fusions, generated by applying multinomial logistic regression, illustrates the potential impact of the *NTRK3* (blue column) and *ETV6* (orange column) fusion genes on surgical decision-making across three different categories: primary total thyroidectomy, not-total thyroidectomy (usually lobectomy), and secondary total thyroidectomy (completion of a not-total thyroidectomy). The likelihoods of the indications for total thyroidectomies (primary or secondary) are represented relative to not-total thyroidectomies (with a baseline value of 0). PTC patients with both *NTRK3* and/or *ETV6* fusion mutations underwent total thyroidectomies more frequently than not-total thyroidectomies. The number of *NTRK3* and/or *ETV6* fusion-positive patients who needed a secondary completion surgery was greater than the number of those with primary total thyroidectomy increasing the risks related to the repeated procedures.

**Figure 5 ijms-25-10787-f005:**
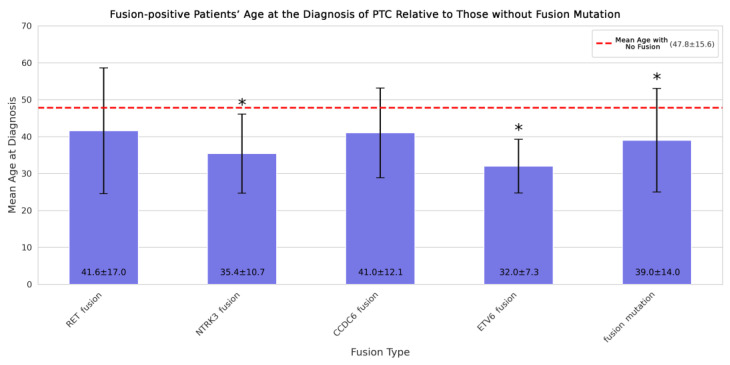
This bar chart illustrates a comparative analysis of the mean age at the diagnosis of PTC between patients carrying those gene fusions occurring at least 6 times in the cohort compared to the mean age of those patients not carrying any gene fusions. The height of the bars along the y-axis represents the mean age of the patients carrying gene fusions. The specific genes are indicated under the corresponding columns with the last column representing an overall positive status for any studied gene fusions (including those mutations with minimal occurrence as well). The red dashed line marks the mean age of the fusion-negative patients. All evaluated fusion mutations were associated with a younger age at the time of diagnosis than the age of patients without any gene fusions, with *NTRK3*, *ETV6*, and general fusion-positive status being significant as marked (*) on the plot. Data are presented as mean ± standard deviation (SD).

**Figure 6 ijms-25-10787-f006:**
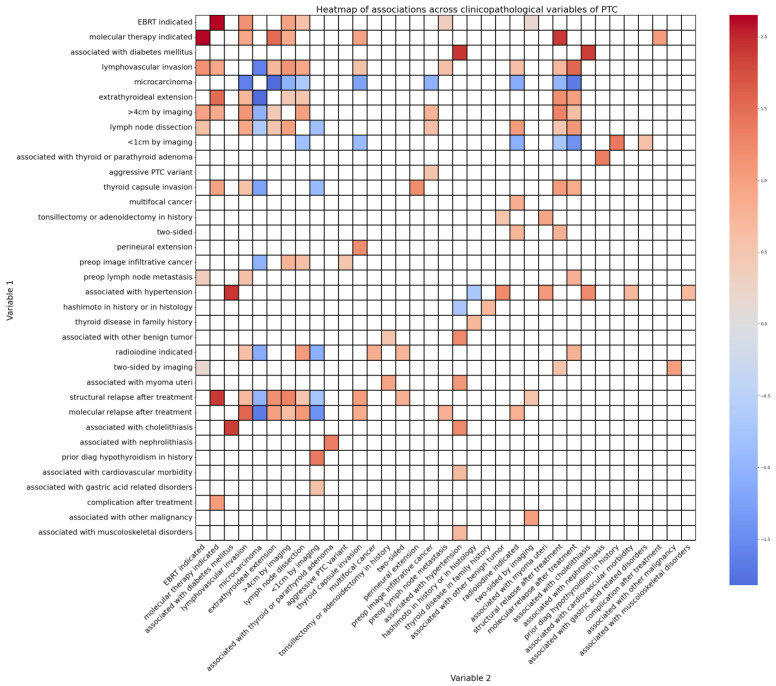
Heatmap listing significant associations between binary-type clinicopathological variables (x-axis and y-axis) of the PTC study cohort. The color scale illustrates the direction of the correlations ranging from strongly positive correlations (red) to strongly negative correlations (blue). Empty (white) cells mark no significant associations. Significant associations mostly tend to occur as clinically expected (e.g., strong positive correlation between lymphovascular invasion and lymph node dissection surgery). Medical indication of molecular therapies explicitly correlated with variables, such as relapse, thyroid capsule invasion, extrathyroidal extension, lymphovascular extension, or need for EBRT, usually related to a more advanced state of illness.

**Table 1 ijms-25-10787-t001:** PTC subtypes in the study cohort.

PTC Subtype	*n* = 100
conventional	69
follicular variant	17
oncocytic	6
tall cell	3
columnar cell	2
encapsulated conventional	1
trabecular	1
Warthin-like	1

**Table 2 ijms-25-10787-t002:** The most relevant malignancies and their associations with fusion genes identified in this study.

Fusion Gene	Cancer Type
RET	thyroid carcinoma, salivary intraductal carcinoma
NTRK3	breast carcinoma, fibrosarcoma
CCDC6	thyroid carcinoma
ETV6	acute lymphoblastic leukemia, breast carcinoma, fibrosarcoma
MET	NSCLC
ALK	anaplastic large T-cell lymphoma
NCOA4	prostate cancer, salivary intraductal carcinoma
EML4	lung cancer

## Data Availability

Some datasets generated and analyzed during the current study are not publicly available but are available from the corresponding author [Ármós R.] on reasonable request.

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
