# Peer review of "Descriptive Analysis of Common Fusion Mutations in Papillary Thyroid Carcinoma in Hungary"

_ijms, 2024, doi:10.3390/ijms251910787_

Round 1

Reviewer 1 Report

Comments and Suggestions for Authors

Comments:

1. Please add patients' information.

2. Please list all abbreviations for readers.

3. What would be the main limitation of the current study? Compared to other similar reports.

4. What would be the findings for treatment of future PTC?

5. If possible, please list of other cancers which are associated with similar fusion mutations. 

6. In Figure 1: please add statistical analysis.

7. In Figure 4: please add statistical analysis. 

8. In Figure 5: what does "*" mean? Compared to what?

Author Response

Thank you for your review. Please see our responses in the attachment.

Reviewer 2 Report

Comments and Suggestions for Authors

Dear Dr Armos

It is a well written paper and the results are very interesting. Just few comments that you can find on the attached file.

Comments on the Quality of English Language

Although I am not a native English speaker, minor editing is required because you use words that have been probably translated from your native language and don't reflect the meaning you want to say in English. (e.g line 61 fight, line 107 can often fall victim, etc.)

Author Response

(The authors gave the same response as above.)

Round 2

Reviewer 1 Report

Comments and Suggestions for Authors

My questions were answered. No more comments.